# Underlying structure and measurement invariance by sex of the state trait anxiety inventory: A psychometric analysis in Ecuador

Jose A. Rodas[1,2]*, Daniel Oleas[3,4], Guido Mascialino[5,6],
José Alejandro Valdevila Figueira[7,8,9], Alberto Rodríguez-Lorenzana[6]

1 Escuela de Psicología, Universidad Espíritu Santo, Samborondón, Ecuador, 2 School of Psychology, University College Dublin, Dublin, Ireland, 3 Department of Research, Universidad Ecotec, Samborondón, Ecuador, 4 Department of Social Anthropology, Basic Psychology and Public Health, Pablo de Olavide University, Seville, Spain, 5 Escuela de Psicología y Educación, Universidad de Las Américas, Quito, Ecuador, 6 Departamento de Ciencias de la Salud, Universidad Pública de Navarra, Pamplona, Navarra, Spain, 7 Escuela de Psicología, Universidad Ecotec, Samborondón, Ecuador, 8 Red de Investigación en Psicología y Psiquiatría (RIPYP), Guayaquil, Ecuador, 9 Instituto de Neurociencias, Junta de Beneficencia de Guayaquil, Guayaquil, Ecuador

* jose.rodas@ucd.ie

## Abstract

Anxiety is currently one of the most prevalent and investigated psychological symptoms requiring precise and reliable instruments for its assessment. The current study evaluates the psychometric properties of the State-Trait Anxiety Inventory (STAI) in an Ecuadorian sample, focusing on the factor structure and potential methodological artefacts, especially those introduced by reverse-scored items. Employing exploratory and confirmatory factor analyses (EFA and CFA), along with structural equation modelling (SEM), the research examines the presence of two distinct factors: genuine anxiety and a secondary factor related to reversed items. Findings indicate that the expected two-factor model fits the data more accurately than a unidimensional model. Notably, cross-loadings suggest the second factor may represent a substantive positive, anxiety-free state rather than solely a methodological effect. Additionally, multigroup CFA confirmed strict measurement invariance across sex, supporting the scale's utility for group comparisons. These results suggest that the STAI's total score represents a composite of anxiety and well-being. While the STAI remains a robust tool for evaluating anxiety in clinical contexts, researchers seeking pure measures of anxiety symptomatology should consider analysing the subscales separately.

## Introduction

### Assessing anxiety

Anxiety has become one of the most common mental health issues worldwide, affecting an estimated 301 million people globally [1]. Systematic reviews such as Baxter

which permits unrestricted use, distribution, and reproduction in any medium, provided the original author and source are credited.

**Data availability statement:** All relevant data are within the manuscript and its Supporting Information files.

**Funding:** University College Dublin funded the publication of the current work. The funders had no role in study design, data collection and analysis, decision to publish, or preparation of the manuscript.

**Competing interests:** All authors declare to have no conflicts of interest. Author JAR reports that University College Dublin funded the publication of the present work. All other authors have nothing to disclose. This does not alter our adherence to PLOS ONE policies on sharing data and materials.

et al. [2] link the existence of this disorder with a significant impact on the quality of life of affected individuals and underline the impact of anxiety not only on emotional well-being but also on daily functioning, work ability and interpersonal relationships. This prevalence underscores the critical need for accurate assessment tools in both clinical and research contexts. Reliable assessment is essential not only for diagnosing disorders and monitoring treatment progress [3] but also for ensuring the validity of research findings and informing public health policy [4–6]. To this end, Spielberger et al. [7] developed the State-Trait Anxiety Inventory (STAI) in 1970. Since its creation, the STAI has become a gold standard for assessing anxiety in clinical contexts [8], evidenced by its widespread use and validation across diverse linguistic and cultural populations, including Spanish [9], French [10], Japanese [11], and Turkish [12] contexts.

### The state-trait anxiety inventory

The STAI allows the assessment of two distinct aspects of anxiety: state anxiety and trait anxiety. The state dimension, measured by the state subscale, identifies transient anxiety, capturing how an individual feels at a given moment and under specific conditions, thereby reflecting fluctuations in anxiety levels [9]. In contrast, the trait dimension is designed to assess an individual's general disposition towards anxiety, independent of situational factors, thus aiming to identify stable patterns of anxious response [14].

Although several investigations have demonstrated the psychometric robustness of the STAI in diverse populations and cultural contexts [13,14] and recent meta-analyses such as Carl et al. [15], reflect the usefulness of the STAI in assessing the effectiveness of psychological and pharmacological interventions for anxiety, there is also empirical evidence that questions the psychometric properties of the instrument [16]. One of the most recurrent criticisms is the possible lack of discriminative ability of the scale between anxiogenic and depressive symptomatology. Studies such as that by [17] suggest that some items included in the STAI may capture aspects of both conditions, and that its specificity may therefore be compromised.

Another point of controversy concerns the factor structure of the STAI, with recent research indicating that the expected unidimensional structure of each of its scales does not consistently replicate across diverse samples and cultural contexts [18,19], raising questions about the validity of both subscales (state and trait anxiety) in all populations as originally theorised. Additionally, it is also possible that item content may not solely measure anxiety but also captures related, albeit distinct, constructs. It appears that, in attempting to assess manifestations of a lack of anxiety, as in the case of positively worded items, other constructs may have inadvertently been introduced into the assessment. Similarly, a significant body of research, using both exploratory and confirmatory factor analyses, supports the presence of two or even three distinct factors underlying responses in both adult and child versions of the STAI [8,20–22].

### The current study

The present study aims to evaluate the psychometric properties of the STAI in an Ecuadorian context, with a focus on addressing the potential impact of methodological

artefacts, particularly reverse-scored items, on its factor structure and construct validity. To achieve this, we employed both exploratory and confirmatory factor analyses (EFA and CFA, respectively), along with structural equation modelling (SEM), to investigate the multi-factor structure of the STAI's state and trait scales.

Given the documented sex differences in the prevalence, expression, and reporting of anxiety symptoms, evaluating measurement invariance by sex is a necessary step in validating the STAI in any population. Women consistently report higher anxiety scores than men across numerous studies [23], which may reflect genuine differences in symptomatology, but could also result from differential item functioning or variation in how the construct of anxiety is measured across sexes. Without establishing that the STAI operates equivalently for men and women—that is, that it measures the same underlying construct with the same structure and meaning—comparisons between groups may be biased or misleading. For these reasons, this study applies multigroup CFA to assess the invariance of the factor structure between male and female participants, who are known to respond differently to anxiety.

In light of the aforementioned concerns, this study seeks to expand the psychometric evaluation of the STAI on the Ecuadorian population, despite the frequent use of anxiety scales in Spanish-speaking populations [e.g., 24–26]. Cultural, linguistic, and contextual factors specific to this population may influence item interpretation and response patterns, making it necessary to confirm that the instrument functions as intended.

We hypothesise that, in line with prior studies and the theoretical position that reverse-scored items introduce systematic method variance rather than a substantive construct: (1) EFA will reveal two correlated factors for both the trait and state questionnaires, largely distinguished by item polarity; (2) a structural model specifying a general anxiety factor (loading on all items) and an uncorrelated method factor (loading exclusively on reversed items) will provide the best fit, confirming that the second factor represents a methodological artefact; (3) both questionnaires will demonstrate strict measurement invariance by sex, indicating that the STAI allows for reliable group comparisons; and (4) female participants will exhibit significantly higher latent anxiety levels than male participants.

## Methods

### Participants

The sample comprised 1,244, residents of Ecuador, ranging in age from 18 to 75 years, with a mean age of 27.65 years (SD = 10.51). Among the participants, 73.31% were single (n = 912), and 26.69% were in a relationship (n = 332). Regarding sex distribution, 61.50% were female (n = 765), and 38.50% were male (n = 479). The majority of the participants were from Guayaquil, accounting for 52.73% of the sample (n = 656), followed by Ambato with 11.17% (n = 139), Babahoyo with 6.51% (n = 81), Quito with 4.10% (n = 51), and 25.48% from other 67 locations across Ecuador (n = 317).

### Instruments

The Spanish version [27] of the State-Trait Anxiety Inventory (STAI) developed by Spielberger et al. (9) was used to assess two independent concepts of anxiety: the emotional state of anxiety (state subscale) and the anxious propensity or anxiety as a personality trait (trait scale). Each subscale features 20 items in a four-point Likert format (ranging from 0 to 3). The state subscale includes 10 reversed items (1, 2, 5, 8, 10, 11, 15, 16, 19, and 20), while the trait subscale contains 7 reversed items (21, 26, 27, 30, 33, 36, and 39). For all analyses, the reversed scores of these items were used.

### Procedure

The study employed a non-probabilistic convenience sampling method, adhering to the following inclusion criteria: 1) being over 18 years of age, 2) residing in Ecuador, and 3) having no prior diagnosis of psychiatric disorders (as determined by a self-report screening question included in the demographic section). Data were collected through a Google form between 16 July 2022 and 30 July 2022. The study was disseminated through official institutional channels, including

university email lists and verified accounts on Facebook, X, and WhatsApp groups associated with the research centre. Data were anonymised, and obtaining written informed consent was mandatory. Prior to participation, all participants were fully informed about the study's objectives and their rights, including the option to withdraw at any time. A specific survey item required participants to provide explicit informed consent before accessing the full survey.

The authors assert that all procedures contributing to this work comply with the ethical standards of the relevant national and institutional committees on human experimentation and with 150 the Helsinki Declaration of 1975, as revised in 2013. All procedures involving human subjects were approved by Comité de Investigación de la Carrera de Psicología de la Universidad Tecnológica Ecotec, approval No 11-27-05-2022.

### Analysis plan

Initial assessments of internal consistency were conducted using McDonald's ω and Cronbach's α. We included ω because it does not assume tau-equivalence, thus providing a more robust reliability estimate for congeneric models. Additional ω indices were estimated after removing single items to identify potentially problematic items.

Exploratory Factor Analysis (EFA) was performed using the Weighted Least Squares (WLS) estimator on a polychoric correlation matrix with oblique rotation (Promax). This approach is appropriate for ordinal data, providing more accurate parameter estimates than Pearson correlations. Sampling adequacy was confirmed via the Kaiser-Meyer-Olkin (KMO) measure and Bartlett's test of sphericity. The optimal number of factors was determined using Parallel Analysis (PA) based on principal components retaining factors with eigenvalues exceeding those generated from random data.

Confirmatory Factor Analysis (CFA) and SEM were subsequently used to determine the nature of the multifactorial solutions. To maximise statistical power for the multigroup invariance analysis—which requires large sample sizes to ensure stable parameter estimates—the full sample was utilised for both EFA and CFA rather than splitting the dataset. Two models were compared for each of the scales: a single-factor model including all items and a two-factor model including a general factor and an uncorrelated 'method' factor consisting exclusively of reversed items. This latter specification was chosen a priori to isolate potential methodological artefacts (e.g., response style) without conflating them with the substantive anxiety construct. The Diagonally Weighted Least Squares (DWLS) estimation method was employed in the lavaan package to account for the ordinal nature of data. The Scaled Chi-Squared Difference Test [28] was used to compare model fit.

Measurement invariance across sex was assessed using a multi-group CFA on the best fitting two-factor model. Given the ordinal nature of the responses, models were estimated using the DWLS estimator with theta parameterisation. This stepwise analysis included testing configural, metric (factor loadings), scalar (item thresholds), and strict (residual) invariance, applying constraints to both the general anxiety and method factor. Theta parameterisation was utilised as it permits the estimation and subsequent constraint of residual variances in ordinal data, allowing for the evaluation of strict invariance. Model comparison relied on changes in fit indices, with values of $\Delta CFI \leq 0.010$ and $\Delta RMSEA \leq 0.015$ serving as the primary criteria for invariance, alongside the traditional $\chi^2$ difference test which is sensitive to sample size. Finally, rather than estimating an additional nested model, latent mean differences across sex were evaluated directly via the Wald test within the strict invariance model.

Model adequacy was evaluated using standard fit indices [29–32]: RMSEA ($\leq 0.08$ acceptable, $\leq 0.05$ good), CFI and TLI ($\geq 0.90$ acceptable, $\geq 0.95$ good), and SRMR ($\leq 0.08$ good). All analyses were performed in R version 4.5.2 [33] and JASP [34], using the lavaan [35], semPlot [36], and semTools [37] packages.

## Results

### Internal consistency

Internal consistency for both the state and trait questionnaires was assessed using McDonald's ω and Cronbach's α. For the state questionnaire, internal consistency was robust, with both metrics providing an identical value of .93, and the

confidence interval was narrow (95% CI [.92 −.93]), indicating excellent reliability. According to Table 1, the ω coefficient remained stable even when individual items were removed from the scale, suggesting that no single item disproportionately influenced the overall consistency.

Similarly, the trait questionnaire demonstrated excellent internal consistency, with both McDonald's ω and Cronbach's α providing a value of.90, and a confidence interval ranging from.90 to.91. As indicated in Table 1, the ω coefficient for the trait questionnaire also did not vary with the exclusion of individual items, underscoring the scale's reliability and the even contribution of items to the consistency measure.

## Factorial structure

An exploratory factor analysis based on a polychoric correlation matrix and employing oblique rotation (Promax method) identified two underlying dimensions within the STAI's State scale (refer to Fig 1). The number of factors was determined via Parallel Analysis, which suggested retaining two components with eigenvalues exceeding those from random data. This two-factor model presents an overall good fit of the data, with all Kaiser-Meyer-Olkin test's indices above.86 and a significant Bartlett's index ($\chi^2(190) = 20943$, $p < .001$). With the exception of item 5, the alignment of items mirrors the anxiety assessment direction—all items comprising the second factor correspond to those reverse-scored. A parallel structure was observed in the Trait scale, where Parallel Analysis also supported a two-factor model presenting a good fit of the data, as evidenced by all Kaiser-Meyer-Olkin values exceeding.92, and a significant Bartlett's test of sphericity ($\chi^2(190) = 16487$, $p < .001$). The items included within the second factor, once more, consist of all items requiring reverse scoring. All factor loadings are presented in Table 2.

**Table 1. ω when individual items are removed from the scale.**

| Item | If item dropped | |
| --- | --- | --- |
| | State | Trait |
| Item 1 | 0.922 | 0.901 |
| Item 2 | 0.921 | 0.896 |
| Item 3 | 0.922 | 0.9 |
| Item 4 | 0.925 | 0.899 |
| Item 5 | 0.921 | 0.898 |
| Item 6 | 0.924 | 0.898 |
| Item 7 | 0.925 | 0.908 |
| Item 8 | 0.928 | 0.897 |
| Item 9 | 0.923 | 0.904 |
| Item 10 | 0.924 | 0.896 |
| Item 11 | 0.926 | 0.897 |
| Item 12 | 0.924 | 0.899 |
| Item 13 | 0.924 | 0.899 |
| Item 14 | 0.923 | 0.901 |
| Item 15 | 0.92 | 0.908 |
| Item 16 | 0.923 | 0.906 |
| Item 17 | 0.923 | 0.902 |
| Item 18 | 0.924 | 0.901 |
| Item 19 | 0.923 | 0.902 |
| Item 20 | 0.922 | 0.902 |

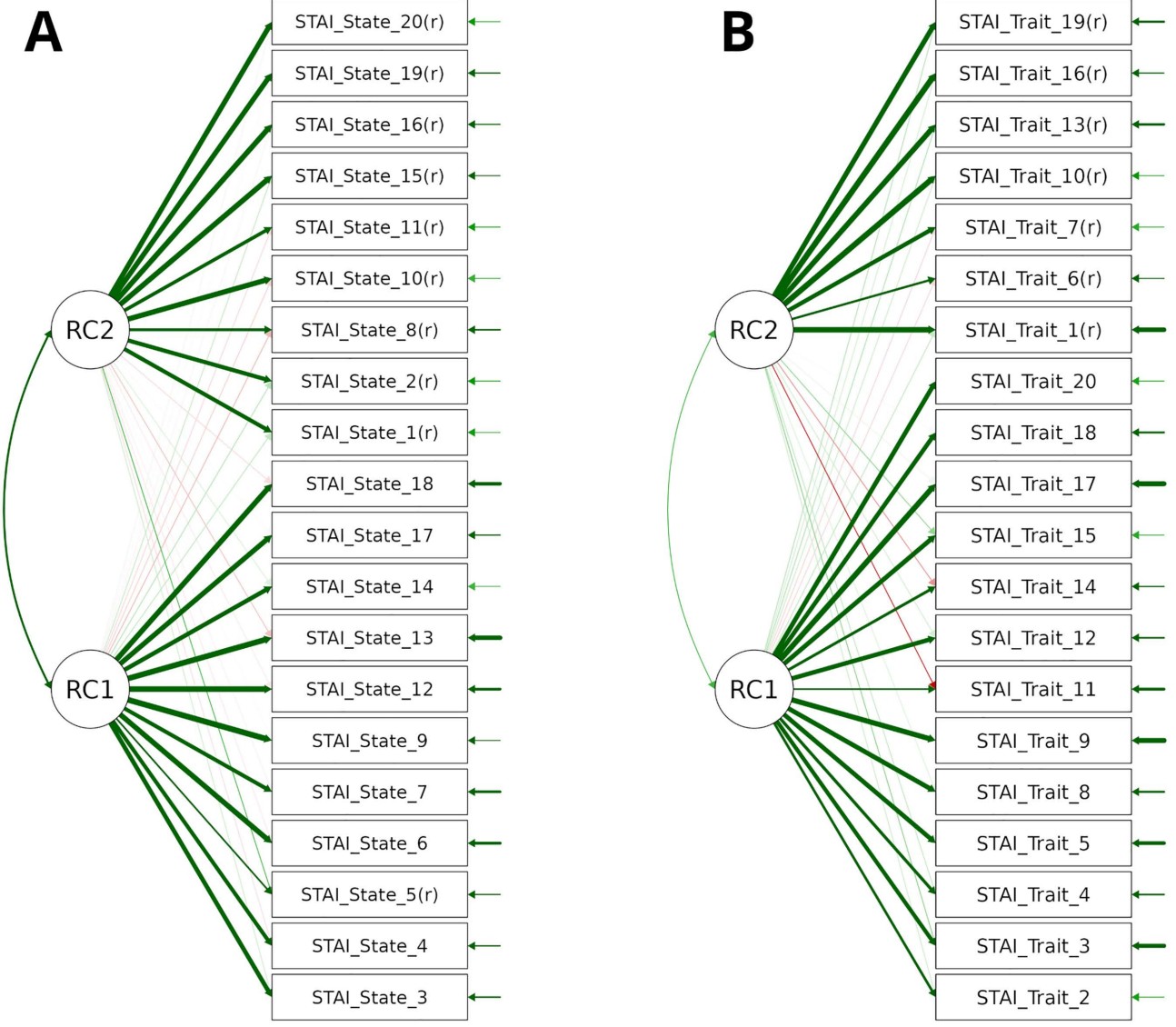

**Fig 1. Two-factor model of the STAI state and trait questionnaires from an exploratory factor analysis.** Panel A presents items from the State questionnaire, while Panel B includes items from the Trait questionnaire. RC1 and RC2 denote the factors within each respective model. The direction of the arrows reflects the hypothesised factor loadings. Arrow thickness indicates the magnitude of these loadings, with thicker arrows denoting stronger relationships between the factors and the observed variables. Arrow colour indicates the sign of the loading: green arrows represent positive loadings, suggesting that higher factor scores are associated with higher item scores, and red arrows indicate negative loadings, where higher factor scores are associated with lower item scores. Arrows with a transparent appearance signify loadings that are not statistically significant.

SEM was used to determine whether a single latent anxiety factor exists and if a secondary factor comprising only reversed items represents a methodological artefact, such as score inversion and response style. This analysis was conducted separately for state and trait anxiety, employing identical procedures for each. Two distinct models were evaluated using the Scaled Chi-Squared Difference Test [28]. The first model consisted of a single factor incorporating all items, whereas the second model consisted of two uncorrelated factors: one representing anxiety, measured as a latent variable incorporating all subscale items, and a methods factor, consisting only of items in reversed order.

**Table 2. Factor loadings from the EFA for state and trait anxiety.**

| | State Questionnaire | | | | Trait Questionnaire | | | |
|---|---|---|---|---|---|---|---|---|
| | | Factor 1 | Factor 2 | Uniqueness | | Factor 1 | Factor 2 | Uniqueness |
| Item 1 | r | | 0.715 | 0.403 | r | | 0.819 | 0.276 |
| Item 2 | r | | 0.731 | 0.367 | | 0.586 | | 0.623 |
| Item 3 | | 0.765 | | 0.365 | | 0.699 | | 0.395 |
| Item 4 | | 0.729 | | 0.502 | | 0.633 | | 0.558 |
| Item 5 | r | 0.525 | | 0.49 | | 0.741 | | 0.45 |
| Item 6 | | 0.831 | | 0.33 | r | | 0.562 | 0.7 |
| Item 7 | | 0.737 | | 0.492 | r | | 0.724 | 0.498 |
| Item 8 | r | | 0.63 | 0.664 | | 0.76 | | 0.396 |
| Item 9 | | 0.878 | | 0.213 | | 0.797 | | 0.364 |
| Item 10 | r | | 0.831 | 0.364 | r | | 0.841 | 0.249 |
| Item 11 | r | | 0.692 | 0.559 | | 0.52 | | 0.667 |
| Item 12 | | 0.858 | | 0.287 | | 0.743 | | 0.411 |
| Item 13 | | 0.875 | | 0.3 | r | | 0.801 | 0.283 |
| Item 14 | | 0.76 | | 0.36 | | 0.616 | | 0.65 |
| Item 15 | r | | 0.823 | 0.231 | | 0.754 | | 0.32 |
| Item 16 | r | | 0.833 | 0.318 | r | | 0.84 | 0.257 |
| Item 17 | | 0.825 | | 0.325 | | 0.816 | | 0.317 |
| Item 18 | | 0.846 | | 0.314 | | 0.741 | | 0.446 |
| Item 19 | r | | 0.808 | 0.351 | r | | 0.774 | 0.336 |
| Item 20 | r | | 0.813 | 0.317 | | 0.751 | | 0.438 |

r = reversed item in the scale, Only loadings above .5 are displayed.

For the State Anxiety questionnaire, the results from the Scaled Chi-Squared Difference Test between the two-factor and one-factor models demonstrated a marked improvement in model fit with the two-factor model ($\Delta\chi^2(10) = 1495.5$, $p < .001$). The one-factor model showed a poor fit, with chi-square values of 10,180.7 (unscaled) and 6,689.3 (scaled), RMSEA = 0.176, CFI = 0.835, TLI = 0.816, and SRMR = 0.168. In contrast, the two-factor model presented a significantly better fit, with a chi-square value of 1,474.1 (scaled = 2388), RMSEA = 0.106, CFI = 0.944, TLI = 0.933, and SRMR = 0.052. Although all items had significant factor loadings, items 8, 10, 11, 16 and 19 had loadings on the general anxiety factor lower than the expected .40 (see Table 3). However, with the exception of item 5 ($\lambda = .24$), these same items exhibited strong loadings on the uncorrelated method factor ($\lambda > .56$). This dissociation supports the interpretation that the variance in these items is largely driven by method effects rather than the core anxiety construct. Modification indices were inspected, revealing substantial residual covariances between specific item pairs, most notably between items 4 and 6 (MI = 555.93). Furthermore, a modification index suggested a cross-loading of Item 3 ('I am tense') onto the method factor (MI = 54.48). However, the standardised expected parameter change was small ($\lambda \approx .10$) and positive, suggesting that any shared variance between this item and the reverse-scored factor is likely artefactual rather than substantive. Consequently, the original model was retained to preserve theoretical parsimony.

Similarly, for the Trait Anxiety questionnaire, the two-factor model again proved superior ($\Delta\chi^2(7) = 1190.9$, $p < .001$). The one-factor model exhibited poor fit, with a chi-square value of 10,473.8 (scaled = 6,226.5), along with RMSEA = 0.169, CFI = 0.789, TLI = 0.765, and SRMR = 0.179. By contrast, the two-factor model demonstrated significantly better adherence to the data characteristics, with a chi-square value of 1,625.3 (scaled = 2,395.8) and improved fit indices: RMSEA = 0.105, CFI = 0.922, TLI = 0.91, and SRMR = 0.073. As in the state questionnaire, all items presented significant factor loadings

**Table 3. Standardised factor loadings from the general and anxiety factor models from CFAs.**

| | State Anxiety | | Trait Anxiety | |
|---|---|---|---|---|
| | **General Factor** | **Anxiety Factor** | **General Factor** | **Anxiety Factor** |
| Item 1 | 0.71 | 0.452 | 0.758 | 0.377 |
| Item 2 | 0.74 | 0.469 | 0.559 | 0.619 |
| Item 3 | 0.765 | 0.823 | 0.761 | 0.81 |
| Item 4 | 0.736 | 0.787 | 0.615 | 0.672 |
| Item 5 | 0.706 | 0.706 | 0.674 | 0.738 |
| Item 6 | 0.828 | 0.869 | 0.362 | 0.119 |
| Item 7 | 0.648 | 0.714 | 0.52 | 0.173 |
| Item 8 | 0.438 | 0.181 | 0.721 | 0.777 |
| Item 9 | 0.836 | 0.882 | 0.738 | 0.791 |
| Item 10 | 0.677 | 0.323 | 0.784 | 0.365 |
| Item 11 | 0.542 | 0.266 | 0.146 | 0.274 |
| Item 12 | 0.79 | 0.838 | 0.711 | 0.769 |
| Item 13 | 0.77 | 0.822 | 0.758 | 0.397 |
| Item 14 | 0.757 | 0.807 | 0.434 | 0.52 |
| Item 15 | 0.806 | 0.488 | 0.804 | 0.852 |
| Item 16 | 0.747 | 0.384 | 0.759 | 0.349 |
| Item 17 | 0.765 | 0.823 | 0.778 | 0.828 |
| Item 18 | 0.77 | 0.823 | 0.692 | 0.745 |
| Item 19 | 0.751 | 0.375 | 0.715 | 0.374 |
| Item 20 | 0.765 | 0.408 | 0.683 | 0.741 |

(see Table 3). However, items 1, 6, 7, 10, 11, 16 and 19 had loadings on the general anxiety factor lower than the expected.40. Consistent with the State scale results, reverse-scored items that displayed weaker associations with the general factor (e.g., item 6r, $\lambda = .12$; item 7r, $\lambda = .17$) showed strong loadings on the method factor ($\lambda = .54$ and $\lambda = .69$, respectively). Inspection of modification indices for the Trait scale revealed a notable tension regarding the orthogonality constraint, particularly involving Item 11 ('I tend to take things too seriously'). The largest index (MI = 816.28) suggested a negative loading for this item on the Method factor. This indicates that the variance captured by the reverse-scored items is not merely methodological but likely includes substantive 'well-being' components. However, we retained the a priori orthogonal structure to avoid conflating construct content with method variance.

## Measurement invariance

Multivariate normality was tested for both questionnaires with the Mardia test. In both cases, normality was not met (non-significant p values) and, due to the ordinal nature of the data, the DWLS method was used for the analyses. Measurement invariance testing was conducted between men and women using a multi-group CFA approach to ensure the scale's applicability for comparing latent constructs of anxiety. The analysis proceeded through stages of configural, metric, scalar, and strict invariance. Initial configural invariance was established, indicating that the factor structure was consistently applicable between men and women (see Table 4). Subsequent testing for metric invariance demonstrated equivalence in factor loadings. Scalar and strict invariance were also supported, showing that both the thresholds and residuals of the items were consistent across groups. Establishing strict invariance is particularly critical, as it confirms that group differences in item scores are due to differences in the latent construct rather than measurement artifacts, thus legitimising the comparison of latent means.

**Table 4. Fit indices for measurement invariance models of the sTAI scales by sex.**

| Scale/ Model | χ2 | df | CFI | RMSEA | Comparison | Δdf | ΔCFI | ΔRMSEA |
|---|---|---|---|---|---|---|---|---|
| **State Anxiety** | | | | | | | | |
| Configural | 1571.12 | 320 | 0.99 | 0.08 | — | — | — | — |
| Metric | 1846.34 | 348 | 0.99 | 0.08 | vs. Configural | 28 | 0 | 0 |
| Scalar | 1697.9 | 368 | 0.99 | 0.08 | vs. Metric | 20 | 0 | −0.01 |
| Strict | 1890.26 | 386 | 0.99 | 0.08 | vs. Scalar | 18 | 0 | 0 |
| **Trait Anxiety** | | | | | | | | |
| Configural | 1812.51 | 326 | 0.98 | 0.09 | — | — | — | — |
| Metric | 1976.6 | 351 | 0.98 | 0.09 | vs. Configural | 25 | 0 | 0 |
| Scalar | 1936.88 | 371 | 0.98 | 0.08 | vs. Metric | 20 | 0 | 0 |
| Strict | 2032.76 | 389 | 0.98 | 0.08 | vs. Scalar | 18 | 0 | 0 |

CFI = Comparative Fit Index; RMSEA = Root Mean Square Error of Approximation. The ΔCFI and ΔRMSEA values represent the absolute change in fit indices relative to the less constrained model. p values correspond to the Satorra-Bentler scaled $\chi^2$ difference test. The "Latent Means" model tests the hypothesis of equal latent means across groups; significant results indicate that latent means differ significantly by sex.

The assessment of latent mean differences, however, indicated significant differences in the constructs' levels across the groups both for state (Standardised Mean Difference = −0.159, p < 0.000) and trait anxiety (Standardised Mean Difference = −0.387, p < 0.000). These results suggest that while the scale measures the construct of anxiety equivalently across groups, there are substantial differences in the levels of anxiety between the groups. Specifically, female participants exhibited significantly higher latent anxiety levels than male participants. Thus, the scale is appropriate for comparative studies that aim to explore and interpret differences in anxiety levels, as it reliably measures the same construct across diverse groups. This confirmation of measurement invariance up to the strict level allows for meaningful comparisons of latent means, thereby providing robust support for the scale's use in cross-group psychological assessments.

## Discussion

The present study evaluated the factor structure of the STAI in an Ecuadorian sample, providing robust evidence for a two-factor model over a unidimensional solution. This aligns with a substantial body of psychometric literature suggesting that the STAI measures two distinct dimensions [8,17]. While the first factor clearly captures genuine anxiety symptoms (e.g., tension, worry), the nature of the second factor—comprising exclusively reverse-scored items—requires more detailed interpretation.

Consistent with previous studies [16,17], we initially hypothesised that this second factor represented methodological noise (e.g., response bias or difficulty processing negations). However, our inspection of modification indices suggests a more substantive interpretation. Notably, in the Trait scale, the anxiety-worded item "I tend to take things too seriously" (Item 11) showed a strong tendency to cross-load negatively onto the reverse-scored factor. If this factor were purely a methodological artefact derived from sentence structure, such content-based cross-loadings would not be expected. Instead, this suggests that the reverse-scored items (e.g., "I feel pleasant," "I feel secure") likely capture a substantive "absence of anxiety" or "well-being" dimension (e.g., carefreeness, relaxation) that is fundamentally incompatible with the cognitive rigidity or worry inherent in taking things too seriously.

This interpretation allows us to contextualise our findings within the broader debate regarding the STAI's structure. One perspective, supported by Bieling et al. [17] and Bados et al. [20], suggests that the STAI conflates anxiety with depression or general negative affect. While our study did not include depression measures to verify convergent validity, the content of the reverse-scored items—reflecting positive affect and emotional stability—overlaps significantly with the absence of depressive symptomatology. A second perspective, proposed by Vigneau & Cormier [8], treats these irregularities

strictly as "method effects." However, our results align most closely with the "Presence vs. Absence of Anxiety" model described by Delgado et al. [38]. In their study of pregnant women, they argued that reverse-scored items form a distinct "absence of anxiety" factor. Our data supports this view: the "method" factor is not merely error variance but represents a coherent psychological state of calmness and well-being that is distinct from the active experience of anxiety.

These findings have significant implications for the use of the STAI in research. The presence of a strong secondary factor suggests that the practice of combining all items into a single total score may obscure important distinctions between "high anxiety" and "low well-being." For researchers aiming to obtain pure measures of anxiety symptomatology, the inclusion of reverse-scored items may introduce variance related to positive affect or social desirability [27]. Therefore, researchers requiring highly specific, unidimensional anxiety measures should consider using alternative scoring methods (e.g., analysing the anxiety-present and anxiety-absent subscales separately) or employing short forms that exclude reversed items.

Despite these structural complexities, our findings support the clinical utility of the STAI. Crucially, we established strict invariance across sexes, indicating that the latent constructs—including the separation of anxiety and method/well-being effects—operate equivalently for men and women. This suggests that while the factor structure is complex, the instrument allows for valid quantitative comparisons of anxiety levels between groups. The instrument remains a valuable tool in clinical settings where ecological validity and the assessment of broad affective distress are often prioritised over factorial purity.

## Limitations and future directions

The results of this study should be interpreted in light of several limitations. First, the cross-sectional design precludes the analysis of test-retest reliability or the stability of the factor structure over time. Second, the study relied on a single measure of anxiety; without concurrent measures of depression (e.g., BDI-II) or positive affect (e.g., PANAS), we could not empirically test convergent and discriminant validity to definitively confirm whether the second factor represents positive affect or lack of depression.

Third, the reliance on online convenience sampling introduces distinct biases. The dissemination of the survey through university networks and social media likely resulted in an overrepresentation of younger, more educated, and digitally literate participants compared to the general Ecuadorian population. Furthermore, online self-report administration can be susceptible to response styles (e.g., rapid responding) that may differ from paper-and-pencil formats used in clinical settings. Fourth, the geographical concentration of the sample in major urban centres, particularly Guayaquil, limits the generalisability of these findings to rural regions or populations with distinct socio-economic backgrounds where cultural interpretations of anxiety may differ. Finally, to maximise statistical power for the multigroup invariance analysis—which requires large sample sizes to ensure stable parameter estimates at the strict invariance level—we opted to utilise the full sample rather than splitting it for separate EFA and CFA cross-validation. While this approach prioritises the robustness of the sex comparison, it limits the immediate external replication of the factor structure within the same dataset.

Future research should address these gaps by incorporating multitrait-multimethod matrices to disentangle anxiety, depression, and method variance in the Ecuadorian population. Additionally, testing bifactor models could help determine if a "general distress" factor accounts for the variance better than the correlated two-factor models used here. Finally, we recommend investigating the psychometric properties of abbreviated versions of the STAI that remove reverse-scored items to determine if they offer a more parsimonious assessment of anxiety for research purposes.

## Conclusion

This study confirms that the STAI is best represented by a two-factor structure in the Ecuadorian context, distinguishing between the presence of anxiety symptoms and a secondary factor composed of reverse-scored items. While often labeled as a "method effect," our analysis suggests this second factor likely captures a substantive dimension of

"well-being" or "anxiety-free" states. Consequently, while the STAI remains a robust and invariant tool for clinical comparison and broad assessment, researchers should be aware that the total score represents a composite of anxiety and well-being. For studies demanding high construct specificity, we recommend caution in interpreting total scores and suggest that separate analyses of the anxiety-present and anxiety-absent items may yield clearer insights into the psychopathology of anxiety.

## Supporting information

**S1 Data. Data frame – STAI.csv. Data frame including all participants.**
(CSV)

## Author contributions

**Conceptualization:** Jose A. Rodas, Daniel Oleas.

**Formal analysis:** Jose A. Rodas, Daniel Oleas.

**Methodology:** Jose A. Rodas, Daniel Oleas.

**Writing – original draft:** Jose A Rodas, Daniel Oleas, Guido Mascialino, José Alejandro Valdevila Figueira, Alberto Rodríguez-Lorenzana.

**Writing – review & editing:** Jose A Rodas, Daniel Oleas, Guido Mascialino, José Alejandro Valdevila Figueira, Alberto Rodríguez-Lorenzana.

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
