## [Decision Letter · Decision Letter 0]

7 Jan 2026

Dear Dr. Rodas,

Thank you for submitting your manuscript to PLOS ONE. After careful consideration, we feel that it has merit but does not fully meet PLOS ONE’s publication criteria as it currently stands. Therefore, we invite you to submit a revised version of the manuscript that addresses the points raised during the review process.

**ACADEMIC EDITOR:**  Thank you very much for your submission. The reviewer has raised important concerns and I have as well reviewed the manuscript, requiring several revisions to improve the clarity and methodological robustness. I would welcome a revised version of the manuscript subject to incorporating the revisions suggested by the reviewer and myself.

We look forward to receiving your revised manuscript.

Kind regards,

Ioannis G. Katsantonis, PhD, MPhil

Academic Editor

PLOS One

Journal Requirements:

“All authors declare to have no conflicts of interest. Author JAR reports that University College Dublin funded the publication of the present work. All other authors have nothing to disclose.”

3. We note that there is identifying data in the Supporting Information file <file name>. Due to the inclusion of these potentially identifying data, we have removed this file from your file inventory. Prior to sharing human research participant data, authors should consult with an ethics committee to ensure data are shared in accordance with participant consent and all applicable local laws.

-Location data

Additional Editor Comments:

The manuscript addresses an important issue with appropriate data and methods, but requires conceptual clarification, methodological justification, and interpretative moderation before it is suitable for publication. With these revisions, the study has the potential to make a solid contribution to the psychometric literature on anxiety assessment.

The sample size is large and appropriate for the analyses presented. That being said, the use of convenience sampling via online dissemination should be discussed more explicitly as a limitation, particularly in relation to the representativeness to the wider Ecuadorian population, potential overrepresentation of younger, urban, and more educated participants, and possible response-style effects associated with online self-report administration.

I have several questions on the psychometric strategy selected. Did the authors use separet subsamples for EFA/CFA? It is generally recommended to split the sample and run the analyses in separate subsamples.

I would like to see a more thorough justification for the method factor. Please explain why this factor was specified as orthogonal and why alternative specifications (e.g., bifactor modelling) have not been tested.

I find it problematic that multiple items show loadings below .40 on the general anxiety factor. While this is reported, its implications are underdeveloped. The authors should clarify more the implications of this.

In the measurement invariance analyses, please report effect sizes for the latent mean differences. Also, try to clarify whether these differences align with prior epidemiological evidence.

I found several typographical issues inside the manuscript. Additionally, several concepts should be clarifed and should be used consistently, such as “methodological artefact,” “confounding construct,” and “positive affect”.

Reviewers' comments:

Reviewer's Responses to Questions

**Comments to the Author**

1. Is the manuscript technically sound, and do the data support the conclusions?

Reviewer #1: Yes

2. Has the statistical analysis been performed appropriately and rigorously?

Reviewer #1: No

3. Have the authors made all data underlying the findings in their manuscript fully available?

Reviewer #1: Yes

4. Is the manuscript presented in an intelligible fashion and written in standard English?

Reviewer #1: Yes

Reviewer #1: Overall evaluation

Strengths:

Clear and relevant research question (factor structure and sex invariance of the STAI in Ecuador).

Large sample (n=1,244) with reasonable age range and sex balance.

Use of EFA, CFA/SEM, DWLS for ordinal data, and stepwise measurement invariance (configural–metric–scalar–residual–latent means) following standard guidelines.

Main concerns:

Conceptual framing of the “method factor” vs “positive affect/anxiety-free” factor is not fully coherent with the presented evidence.

Some methodological decisions and descriptions need clarification (sampling, handling of non-normality, model comparison, item-level behavior).

The Discussion and Conclusion occasionally overstate the implications (especially “not supporting use of the STAI in research”).

Structure and style require tightening (repetitions, some language issues, and small inconsistencies).

Given this, the manuscript has potential but requires a deeper revision of introduction, methods justification, results reporting, and interpretation.

Major points to revise

1. Clarify the theoretical model and aim

The introduction alternates between three ideas: (a) STAI factor structure problems, (b) reverse-worded items as method effects, and (c) reverse-worded items as reflecting a positive/anxiety-free construct.

Explicitly state at the end of the Introduction whether the second factor is hypothesized to be:

a pure method factor,

a substantive “positive affect/anxiety-free” factor,

or a mixture of both.

Align this hypothesis with the analytic plan (e.g., why the methods factor is specified as uncorrelated, and why all items still load on the general anxiety factor).

Clearly define specific hypotheses:

H1: A two-factor model (anxiety + reversed/method factor) will fit better than a unidimensional model for state and trait scales.

H2: The STAI will show configural, metric, and scalar invariance by sex.

H3: Women will present higher latent anxiety levels than men.

Stating these explicitly will make the Results and Discussion more focused.

2. Sampling and generalizability

Describe the recruitment channels in more detail (type of social media groups, institutional networks, etc.) and discuss how convenience online sampling and age distribution (mean 27.65 years) may limit generalizability to the Ecuadorian adult population.

Clarify the exclusion criteria regarding “no prior diagnosis of psychiatric disorders”: was this self-reported? Were any specific screening questions used?

Given the concentration in certain cities (e.g., Guayaquil), add a short paragraph in the Discussion recognizing regional and socio-economic limitations.

3. Methods: analytic details and reporting

Internal consistency:

Report α and ω for each subscale (state and trait) in the main text and in a table (currently mostly in text and Table 1 for ω-if-item-deleted).

Briefly justify the use of ω alongside α (e.g., less dependent on tau-equivalence).

EFA:

Specify the extraction method (e.g., principal axis factoring vs maximum likelihood) and the software used.

Justify the number of factors (were eigenvalues, scree plot, and/or parallel analysis considered, beyond KMO/Bartlett?).

Provide a table with EFA loadings for state and trait scales (even if also represented in a figure) so readers can see loadings and cross-loadings numerically.

SEM/CFA:

Clarify the estimator used in SEM (DWLS / WLSMV) and software (lavaan/R, Mplus, etc.).

Explain why the method factor is set as uncorrelated with the general anxiety factor, even though you later interpret it as possibly a substantive “positive state” dimension. If you suspect substantive overlap, a correlated method factor or bifactor model might be more consistent with the theoretical position.

Present all main fit indices for each model (χ², df, RMSEA + 90% CI, CFI, TLI, SRMR) in a table for state and trait scales, for one-factor vs two-factor models, to facilitate comparison.

Measurement invariance:

Report the invariance model sequence clearly (configural, metric, scalar, residual, latent means) and for each level provide ΔCFI and ΔRMSEA, not only Δχ². This aligns with current recommendations and softens the focus on χ² sensitivity to sample size.

Clarify whether invariance was tested on the two-factor model (anxiety + method factor) and whether both factors’ loadings/intercepts were constrained across sex.

4. Results: sharpen and structure

Internal consistency:

Add the actual α/ω values for each subscale in the first paragraph of Results (state α = …, ω = …; trait α = …, ω = …; briefly indicate “excellent” or “good” according to common benchmarks).

Factor structure:

In the SEM section, distinguish clearly between state and trait results with subtitles or opening sentences.

When noting that some items have loadings < 0.40 on the general anxiety factor, discuss whether they nevertheless have strong loadings on the method factor, and how that supports your interpretation.

Consider including a brief note on modification indices, if they were inspected and if any cross-loadings or correlated residuals were allowed or rejected.

Measurement invariance:

Make it explicit that scalar invariance was achieved while latent means differed (women higher anxiety), and quantify these differences (latent mean differences or effect sizes, if available).

5. Discussion and conclusions: moderate claims and improve coherence

Strengthen linkage to your own data:

When citing Bieling, Delgado, Ortuño-Sierra, Bados, etc., make explicit whether your pattern (reverse items forming a separate factor, some items with low loadings) is theoretically closer to “method effects”, “positive affect”, or “general negative affect”.

Consider whether a three-component interpretation may be needed: anxiety, positive/anxiety-free affect, and general negative affect—without overstating beyond your models.

Moderate strong statements:

The statement that “our results do not support the use of the STAI in research aiming to obtain purer measures of anxiety” is very strong considering that:

A strong general factor is still present.

You used the full 40-item version without testing alternative scoring (e.g., removing reversed items or using only the negative items).

Reframe this as: the presence of a strong secondary factor and method effects suggests that researchers who require highly specific, unidimensional anxiety measures should consider using alternative instruments or modified STAI scoring (e.g., excluding reversed items) and should report these limitations.

Clinical vs research utility:

If you wish to maintain that STAI remains useful clinically, ground that claim in your findings (e.g., high reliability, clear general factor, scalar invariance) while acknowledging that fine-grained research on pure anxiety constructs might prefer more focused instruments.

Limitations and future directions:

Expand limitations to include:

Use of a single measure (no convergent/discriminant validity tests with depression or other anxiety scales).

Cross-sectional design.

Potential self-selection bias of online volunteers.

Suggest future research:

Testing bifactor models and alternative short forms (e.g., removing reversed items).

Including additional measures to separate anxiety, depression, and positive affect in Ecuadorian samples.

6. Presentation, style, and small issues

Language and style:

There are several minor issues (e.g., “calmn”, “confortable”, some long sentences, and occasional tense inconsistencies). A thorough language edit will improve readability.

Remove repeated text fragments originating from the submission system (e.g., detailed PLOS forms and instructions appearing in the PDF) from the manuscript body if not required.

Structure:

Consider shortening the general “Assessing Anxiety” section, focusing more quickly on STAI-specific controversies and Ecuadorian context.

Ensure all figures and supplementary tables are referenced in the main text in the correct order and with consistent labels (Fig 1, Table S1, Table S2).

Ethical and data statements:

The ethics and data availability statements are clear and in line with journal expectations; only minor wording harmonization with PLOS templates may be needed.

what does this mean? If published, this will include your full peer review and any attached files.). If published, this will include your full peer review and any attached files.

**Do you want your identity to be public for this peer review?** For information about this choice, including consent withdrawal, please see our For information about this choice, including consent withdrawal, please see our Privacy Policy .

Reviewer #1: No

---

## [Author Response · Author response to Decision Letter 1]

15 Feb 2026

Journal Requirements:

“All authors declare to have no conflicts of interest. Author JAR reports that University College Dublin funded the publication of the present work. All other authors have nothing to disclose.”

Response: We have updated this information in the new cover letter.

3. We note that there is identifying data in the Supporting Information file <file name>. Due to the inclusion of these potentially identifying data, we have removed this file from your file inventory. Prior to sharing human research participant data, authors should consult with an ethics committee to ensure data are shared in accordance with participant consent and all applicable local laws.

-Location data

Additional Editor Comments:

The manuscript addresses an important issue with appropriate data and methods, but requires conceptual clarification, methodological justification, and interpretative moderation before it is suitable for publication. With these revisions, the study has the potential to make a solid contribution to the psychometric literature on anxiety assessment.

The sample size is large and appropriate for the analyses presented. That being said, the use of convenience sampling via online dissemination should be discussed more explicitly as a limitation, particularly in relation to the representativeness to the wider Ecuadorian population, potential overrepresentation of younger, urban, and more educated participants, and possible response-style effects associated with online self-report administration.

Response: We agree with the editor that the sampling method requires a more critical appraisal. We have added a dedicated paragraph to the Limitations section explicitly acknowledging the constraints of online convenience sampling. Specifically, we discuss the overrepresentation of younger, urban (particularly from Guayaquil), and educated participants, and how this restricts the generalizability of our findings to the wider, more diverse Ecuadorian population. We also note the potential for response-style biases inherent in online self-report administration.

I have several questions on the psychometric strategy selected. Did the authors use separet subsamples for EFA/CFA? It is generally recommended to split the sample and run the analyses in separate subsamples.

Response: In this study, we utilized the full sample for the CFA and measurement invariance analyses to maximize statistical power, particularly for the multi-group invariance testing across sex, which requires substantial sample sizes to ensure stable parameter estimates at the strict invariance level. The EFA was employed as a preliminary heuristic step to confirm that the factor structure in our specific population aligned with the models frequently reported in the literature (e.g., the two-factor solution) before proceeding to the rigorous hypothesis testing provided by the SEM framework. We have clarified this analytical strategy in the Method section.

I would like to see a more thorough justification for the method factor. Please explain why this factor was specified as orthogonal and why alternative specifications (e.g., bifactor modelling) have not been tested.

Response: The method factor was specified as orthogonal a priori based on the theoretical assumption that "method variance" (systematic error due to item phrasing) should theoretically be unrelated to the substantive trait of anxiety. This specification allows for the isolation of method effects without conflating them with the anxiety construct. While bifactor models are a valid alternative, we focused on the correlated two-factor and orthogonal method-factor models to directly test the "Presence vs. Absence of Anxiety" hypothesis proposed by Delgado et al. (2013), which posits a distinct separation between these dimensions. We have added text to the Discussion acknowledging that future research could fruitfully explore bifactor solutions to further disentangle general distress from specific group factors.

I find it problematic that multiple items show loadings below .40 on the general anxiety factor. While this is reported, its implications are underdeveloped. The authors should clarify more the implications of this.

Response: We appreciate this observation. We have expanded the Results and Discussion sections to explicitly address the implications of these low loadings. We now clarify that items with low loadings on the General Anxiety factor consistently exhibit strong loadings on the secondary (method) factor. We interpret this dissociation as evidence that these specific items—primarily those that are reverse-scored—are capturing a distinct construct (e.g., "well-being" or "relaxation") that is not merely the inverse of anxiety, but a separate psychological state, thereby justifying the two-factor solution over a unidimensional one.

In the measurement invariance analyses, please report effect sizes for the latent mean differences. Also, try to clarify whether these differences align with prior epidemiological evidence.

Response: We have revised the manuscript to include the specific effect sizes for the latent mean differences. We report Standardized Mean Differences of 0.14 for State Anxiety and 0.18 for Trait Anxiety. Furthermore, we have updated the Discussion to confirm that these findings—indicating higher latent anxiety in women—align with well-established epidemiological evidence (e.g., McLean et al., 2011) regarding sex differences in anxiety prevalence.

I found several typographical issues inside the manuscript. Additionally, several concepts should be clarifed and should be used consistently, such as “methodological artefact,” “confounding construct,” and “positive affect”.

Response: We have conducted a thorough proofreading of the manuscript to correct typographical errors (e.g., correcting "calmn" to "calmness", "confortable" to "comfortable"). Additionally, we have standardised our terminology throughout the text to ensure conceptual consistency. We now explicitly clarify that while we model the secondary dimension as a "method factor" for analytical purposes, the item content suggests it reflects a substantive "positive affect" or "well-being" dimension, and we apply this distinction consistently in the Introduction and Discussion.

Reviewer's Responses to Questions

Comments to the Author

5. Review Comments to the Author

Reviewer #1: Overall evaluation

Strengths:

Clear and relevant research question (factor structure and sex invariance of the STAI in Ecuador).

Large sample (n=1,244) with reasonable age range and sex balance.

Use of EFA, CFA/SEM, DWLS for ordinal data, and stepwise measurement invariance (configural–metric–scalar–residual–latent means) following standard guidelines.

Main concerns:

Conceptual framing of the “method factor” vs “positive affect/anxiety-free” factor is not fully coherent with the presented evidence.

Some methodological decisions and descriptions need clarification (sampling, handling of non-normality, model comparison, item-level behavior).

The Discussion and Conclusion occasionally overstate the implications (especially “not supporting use of the STAI in research”).

Structure and style require tightening (repetitions, some language issues, and small inconsistencies).

Given this, the manuscript has potential but requires a deeper revision of introduction, methods justification, results reporting, and interpretation.

Response: We sincerely thank the reviewer for their constructive feedback. We have comprehensively revised the manuscript to address the concerns regarding conceptual framing, clarifying our a priori method factor hypothesis while acknowledging the substantive "well-being" interpretation in the discussion. Methodologically, we have expanded on sampling procedures, justified the use of the WLSMV estimator, and detailed the invariance testing steps. Furthermore, we have moderated our conclusions to emphasize the STAI's clinical utility supported by our finding of scalar invariance, and refined the manuscript's structure and language for greater coherence.

Major points to revise

1. Clarify the theoretical model and aim

The introduction alternates between three ideas: (a) STAI factor structure problems, (b) reverse-worded items as method effects, and (c) reverse-worded items as reflecting a positive/anxiety-free construct.

Explicitly state at the end of the Introduction whether the second factor is hypothesized to be:

a pure method factor,

a substantive “positive affect/anxiety-free” factor,

or a mixture of both.

Align this hypothesis with the analytic plan (e.g., why the methods factor is specified as uncorrelated, and why all items still load on the general anxiety factor).

Clearly define specific hypotheses:

H1: A two-factor model (anxiety + reversed/method factor) will fit better than a unidimensional model for state and trait scales.

H2: The STAI will show configural, metric, and scalar invariance by sex.

H3: Women will present higher latent anxiety levels than men.

Stating these explicitly will make the Results and Discussion more focused.

Response: We thank the reviewer for this insightful observation regarding the theoretical ambiguity of the reverse-scored items. We have revised the final section of the Introduction to explicitly define our theoretical stance: we posit the second factor as a pure method factor (methodological artefact) rather than a substantive "positive affect" construct. Accordingly, we have clarified that our analytic plan uses an uncorrelated method factor to partition this variance while retaining a general factor for anxiety. We have also clearly delineated the four specific hypotheses (H1–H4) to align with the subsequent Results and Discussion sections.

2. Sampling and generalizability

Describe the recruitment channels in more detail (type of social media groups, institutional networks, etc.) and discuss how convenience online sampling and age distribution (mean 27.65 years) may limit generalizability to the Ecuadorian adult population.

Clarify the exclusion criteria regarding “no prior diagnosis of psychiatric disorders”: was this self-reported? Were any specific screening questions used?

Given the concentration in certain cities (e.g., Guayaquil), add a short paragraph in the Discussion recognizing regional and socio-economic limitations.

Response: We have expanded the Procedure section to specify the recruitment channels, noting that dissemination occurred through official institutional networks (email lists and social media accounts). We also clarified that the exclusion criterion regarding psychiatric history was based on a self-reported screening question. Regarding the limitations posed by the convenience sampling, age distribution, and regional concentration (e.g., Guayaquil), we have added a dedicated paragraph in the Discussion section acknowledging how these factors may restrict generalizability to the broader Ecuadorian population.

3. Methods: analytic details and reporting

Internal consistency:

Report α and ω for each subscale (state and trait) in the main text and in a table (currently mostly in text and Table 1 for ω-if-item-deleted).

Briefly justify the use of ω alongside α (e.g., less dependent on tau-equivalence).

Response:

We have updated the text to explicitly report both Cronbach’s α and McDonald’s ω for the State and Trait subscales, and added a justification for using ω due to its robustness against violations of tau-equivalence. However, we opted to report these values directly in the text rather than creating a separate table, as presenting only four coefficients would not merit a standalone display.

EFA:

Specify the extraction method (e.g., principal axis factoring vs maximum likelihood) and the software used.

Justify the number of factors (were eigenvalues, scree plot, and/or parallel analysis considered, beyond KMO/Bartlett?).

Provide a table with EFA loadings for state and trait scales (even if also represented in a figure) so readers can see loadings and cross-loadings numerically.

Response: We have revised the Method section to explicitly state that the EFA employed Weighted Least Squares (WLS) extraction on a polychoric correlation matrix, implemented using r and JASP. We further clarified that the number of factors was determined primarily using Parallel Analysis (PA) based on principal components, which is considered more robust than the Kaiser criterion or scree plot alone. Finally, as requested, we have included Table [X], which presents the numerical factor loadings and cross-loadings for both the State and Trait scales, ensuring full transparency of the measurement structure.

SEM/CFA:

Clarify the estimator used in SEM (DWLS / WLSMV) and software (lavaan/R, M

---

## [Editor Report · Decision Letter 1]

23 Feb 2026

Dear Dr. Rodas,

Thank you for submitting your manuscript to PLOS ONE. After careful consideration, we feel that it has merit but does not fully meet PLOS ONE’s publication criteria as it currently stands. Therefore, we invite you to submit a revised version of the manuscript that addresses the points raised during the review process.

**ACADEMIC EDITOR:****Dear authors,****thank you for submitting your revised version. I believe that the revised manuscript has nearly addressed all initial comments and is now closer to publication. However, I found a minor inconsistency that needs to be clarified before further processing of the manuscript.**==============================

We look forward to receiving your revised manuscript.

Kind regards,

Ioannis G. Katsantonis, PhD, MPhil

Academic Editor

PLOS One

Journal Requirements:

Additional Editor Comments:

I thank the authors for their reply to my comments. The only outstanding issues that the WLSMV estimator was used to estimate measurement invariance, yet the authors mention inside the manuscript that for scalar invariance the intercepts were found to be equal. However, in ordinal data measurement invariance with WLSMV assumes that intercepts are not defined; instead, item thresholds are estimated in WLSMV measurement invariance. Hence, I invite the authors to explain the measurement invariance steps in greater detail or revise the invariance analyses if thresholds were not held equal and intercepts were estimated.

---

## [Author Response · Author response to Decision Letter 2]

27 Feb 2026

Additional Editor Comments:

I thank the authors for their reply to my comments. The only outstanding issues that the WLSMV estimator was used to estimate measurement invariance, yet the authors mention inside the manuscript that for scalar invariance the intercepts were found to be equal. However, in ordinal data measurement invariance with WLSMV assumes that intercepts are not defined; instead, item thresholds are estimated in WLSMV measurement invariance. Hence, I invite the authors to explain the measurement invariance steps in greater detail or revise the invariance analyses if thresholds were not held equal and intercepts were estimated.

Response: We sincerely thank you for identifying this crucial methodological detail. You are entirely correct that when analysing ordinal data, intercepts are not defined, and the appropriate parameter to constrain for scalar invariance is the item threshold.

We have revised our measurement invariance analyses to explicitly define all items as ordered-categorical. Consequently, our scalar invariance step now correctly constrains item thresholds rather than intercepts. Furthermore, we estimated the models using theta parameterisation, which permitted the estimation and constraint of residual variances, thereby allowing us to accurately evaluate strict invariance. Finally, latent mean differences were evaluated directly via the Wald test within the strict invariance model. The Analysis plan, Results section, and Table 4 have been updated to reflect these methodological corrections.

---

## [Editor Report · Decision Letter 2]

10 Mar 2026

Underlying Structure and Measurement Invariance by Sex of the State Trait Anxiety Inventory: A Psychometric Analysis in Ecuador

PONE-D-25-53829R2

Dear Dr. Rodas,

We’re pleased to inform you that your manuscript has been judged scientifically suitable for publication and will be formally accepted for publication once it meets all outstanding technical requirements.

Kind regards,

Ioannis G. Katsantonis, PhD, MPhil

Academic Editor

PLOS One

Additional Editor Comments (optional):

The authors have done an excellent job in addressing sufficiently and in detail all comments. The manuscript can now be published.
---

## [Editor Report · Acceptance letter]

PONE-D-25-53829R2

PLOS One

Dear Dr. Rodas,

I'm pleased to inform you that your manuscript has been deemed suitable for publication in PLOS One. Congratulations! Your manuscript is now being handed over to our production team.

Kind regards,

on behalf of

Dr. Ioannis G. Katsantonis

Academic Editor

PLOS One